# The Who, When, Why, and How of PET Amyloid Imaging in Management of Alzheimer’s Disease—Review of Literature and Interesting Images

**DOI:** 10.3390/diagnostics9020065

**Published:** 2019-06-25

**Authors:** Subapriya Suppiah, Mellanie-Anne Didier, Sobhan Vinjamuri

**Affiliations:** 1Centre for Diagnostic Nuclear Imaging, University Putra Malaysia, Serdang 43400, Selangor, Malaysia; subapriya@upm.edu.my; 2Department of Imaging, Faculty of Medicine and Health Sciences, University Putra Malaysia, Serdang 43400, Selangor, Malaysia; 3The Royal Liverpool and Broadgreen University Hospitals NHS Trusts, Prescot St, Liverpool L7 8XP, UK; mellanie-anne.didier@rlbuht.nhs.uk; 4Section of Nuclear Medicine, Department of Surgery, Radiology, Anaesthesia & Intensive Care, The University Hospital of The West Indies, The University of The West Indies, Mona Campus, Kingston 7, Jamaica

**Keywords:** Alzheimer’s disease, diagnostic imaging, molecular imaging, precision medicine, quantification, nuclear medicine, ^18^F-FDG, PET, neurocognitive disorder

## Abstract

Amyloid imaging using positron emission tomography (PET) has an emerging role in the management of Alzheimer’s disease (AD). The basis of this imaging is grounded on the fact that the hallmark of AD is the histological detection of beta amyloid plaques (Aβ) at post mortem autopsy. Currently, there are three FDA approved amyloid radiotracers used in clinical practice. This review aims to take the readers through the array of various indications for performing amyloid PET imaging in the management of AD, particularly using 18F-labelled radiopharmaceuticals. We elaborate on PET amyloid scan interpretation techniques, their limitations and potential improved specificity provided by interpretation done in tandem with genetic data such as apolipiprotein E (APO) 4 carrier status in sporadic cases and molecular information (e.g., cerebral spinal fluid (CSF) amyloid levels). We also describe the quantification methods such as the standard uptake value ratio (SUVr) method that utilizes various cutoff points for improved accuracy of diagnosing AD, such as a threshold of 1.122 (area under the curve 0.894), which has a sensitivity of 92.3% and specificity of 90.5%, whereas the cutoff points may be higher in APOE ε4 carriers (1.489) compared to non-carriers (1.313). Additionally, recommendations for future developments in this field are also provided.

## 1. Introduction and Role of [^18^F]FDG PET

Diagnostic imaging has always played an important role in the management of Alzheimer’s disease (AD). Conventionally, neurologists and psychiatrists depended on magnetic resonance imaging (MRI) to help diagnose probable AD, particularly in atypical cases, by identifying typical anatomical changes that are characteristic of AD [1]. This type of structural imaging, however, has its limitations due to naturally occurring variations in brain volume caused by the aging process and co-existing conditions that can cause medial temporal lobe atrophy, such as depression [2]. Hence, functional imaging such as positron emission tomography (PET) offers an irresistible enticement for increased accuracy of diagnosis. PET is very important in providing a one-stop solution and its function is not limited only to the oncology field [3].

There are two main categories of radiopharmaceuticals that are utilized for PET/CT imaging in patients suspected with AD. Initially, 2-Deoxy-2-[^18^F]fluorodeoxyglucose ([^18^F]FDG), a glucose analog, was utilized for PET brain imaging for the management of AD. Healthy brain cells avidly take up the substance, as they highly metabolize glucose, but the substance is relatively reduced in uptake in the temporo-parietal cortical regions that are affected by AD. Although the role of [^18^F]FDG has been established for making the diagnosis of AD, the accuracy of the scan interpretations can decline markedly when it involves older patients. In a younger cohort with a mean age of 64 years of age, the sensitivity of [^18^F]FDG PET was reported to be 100%, with a specificity of 75% (accuracy 84%), whereas a study with older patients has reported 20% lower accuracy in late-onset AD [4]. Consequently, this limitation was the catalyst for the development of more specific biomarkers for the detection of AD, namely amyloid precursors. The detection of amyloid precursors is said to be able to predict the conversion of at risk subjects to full blown AD 10 years prior to the onset of AD symptoms [5].

Amyloid precursors for PET imaging such as the short radioactive half-life (20 minutes) Pittsburgh compound B ([^11^C]PiB), by crossing the blood brain barrier (BBB) and selectively binding to beta amyloid plaques (Aβ), are able to provide a virtual ante mortem histopathological portrait of the brain. Similar but newer radiopharmaceuticals include ^18^F-labeled Aβ targeting tracers, such as [^18^F]Florbetapir (Amyvid, Eli Lilly, USA) ([^18^F]FBP), [^18^F]Florbetaben (Neuraceq, Piramal, Mumbai) ([^18^F]FBB), and [^18^F]Flutemetamol (Vizamyl GE Healthcare, USA) ([^18^F]FMT) [6,7,8,9]. These amyloid radiotracers are more suited for clinical settings and have high specificity for the detection of Aβ plaques [10]. These are relatively novel radiopharmaceuticals used in the management of AD as they also enable the quantification of Aβ plaque burden in the brain cortices, which is a hallmark of AD.

Previous research has focused on [^18^F]FDG PET studies as it is a widely available radiotracer for various imaging indications and has well established cutoff points for standardized uptake values (SUVs) utilized for disease process quantification. More specifically, [^18^F]FDG PET is used to differentiate AD from other clinical diagnoses by normalizing the uptake intensity to the mean metabolic rate for glucose utilization in the whole brain (CMR_glc_) or the cerebellar glucose consumption, as these areas allow for accurate distinction of AD, by being maximally stable in subjects but minimally affected by external stimuli and are relatively unaffected by the disease of interest [11]. Nevertheless, many factors influence the value of the measured glucose uptake, namely patient related factors such as fasting blood glucose levels and altered bio-distribution of [^18^F]FDG [12]. Additionally, an inverse relationship has been noted between cortical retention of amyloid compared with cerebral glucose metabolism determined with [^18^F]FDG, which was detected to be most robust in the posterior temporoparietal lobes [13]. This pathognomonic finding on [^18^F]FDG PET, however, may be absent in certain cases and regional fibrillary amyloid depositions have been noted to have little to nil significant association with regional cortical FDG hypometabolism, but rather the impaired FDG metabolism is more influenced by global amyloid burden [14].

AD, which is a major neurocognitive disorder (NCD), was first described by Alois Alzheimer in 1906, by discovering intracerebral neuritic plaques and neurofibrillary tangles (NFTs) in the post mortem study of a patient with chronic mental illness [15]. AD is currently the most common NCD, with more than 36 million cases worldwide and nearly catastrophic in its prevalence among older adults [16]. As a matter of fact, the prevalence of AD after the age of 85 years old is 85 per 1000 population. Approximately 1% of AD is autosomal dominant inheritance due to mutations on chromosomes 21 (APP), 14 (presenilin 1), and 1 (presenilin 2), which leads to early onset familial Alzheimer’s disease (oeFAD). Nevertheless, most cases are idiopathic and often the diagnosis is elusive until a later stage in the disease progression. There are several subtypes of neurocognitive disorders (NCDs) which include Alzheimer’s disease (AD), dementia with Lewy bodies (DLB), vascular dementia (VaD), and fronto-temporal dementia (FTD). Some sufferers have mixed type of NCDs; however, AD is the commonest subtype with over 60% of subjects suffering from this disorder.

Diagnosis of AD is made primarily on the basis of clinical criteria using the Diagnostic and Statistical Manual, 4th and 5th editions (DSM-IV and DSM-5) [17]. This method is combined with neuropsychiatric testing to objectively assess for cognitive impairment, i.e., Mini Mental State Examination (MMSE) or the Montreal Cognitive Assessment (MoCA) scoring [18]. For instance, in the absence of secondary causes, using MMSE scoring helps to grade the severity of cognitive decline with subjects scoring < 15/30 as having severe AD, < 24/30 being more likely to have mild AD, as well as subjects with reported memory deficits, and normal or slightly low scores but preserved ADL (activities of daily living), of having mild cognitive impairment (MCI) [17]. Moreover, subject education level has to be factored in when interpreting the assessment score. A set of standards created in 1984 by the National Institute of Neurological and Communicative Disorders and Stroke (NINCDS) and the Alzheimer’s disease and Related Disorders Association (ADRDA) known as the NINCDS-ADRDA criteria had a sensitivity of 81% and specificity of 70% of diagnosing AD [19]. However, due to new developments in the field, Knopman et al., proposed recommendations in 2001 [20], which were later revised in 2011 [21].

The National Institute on Aging-Alzheimer’s Association (NIA-AA) workgroups involved in preparation of diagnostic guidelines for AD, set a revised guideline that took into account distinguishing features of other dementing/neurocognitive deficit causing conditions that occur in a similarly aged population, which were not completely recognized in the past [21]. These criteria could help differentiate NCD subtypes such as DLB, VaD, and FTD from AD, as they have inherently different clinical characteristics, as well as unique pathophysiological features. The criteria propose the use of two classes of biomarkers to help diagnose AD, namely biomarkers for brain amyloid Aβ protein deposition detection and the biomarkers for downstream neuronal injury identification [21]. The former class includes two major biomarkers namely detection of reduced cerebrospinal fluid (CSF) Aβ_42_ levels and positive PET amyloid imaging, whereas the latter includes three major biomarkers, namely elevated CSF tau, reduced [^18^F]FDG uptake on PET in the temporoparietal lobes, and disproportionate atrophy at the medial, basal, and lateral temporal lobes on structural MRI scans [21]. Nevertheless, the criteria have their limitations with regards to absolute classification of subtypes due to the presence of certain inherent heterogeneity within the groups. For example, it is difficult to differentiate NCD subjects who have tauopathy disorders such as FTD, progressive supranuclear palsy (PSP), and cortico-basal degeneration [22].

In addition, the International Working Group (IWG) and the US NIA-AA have contributed criteria for the diagnosis of AD that better define clinical phenotypes and integrate the role of neuroimaging, genetic profiling, and CSF biomarkers [23]. Some of the recommendations are stated as a guideline and others are given as an option to be utilized at the discretion of the clinician. At present it is accepted that a good clinical assessment yields fairly good sensitivity and specificity to diagnose the different subtypes of NCDs and is reasonably on par with other types of investigations to diagnose AD. The IWG has proposed revisions to the diagnostic algorithm for defining typical and atypical AD, identifying mixed type of AD and preclinical states of AD [23].

The objective of this review is to expound on the various arrays of indications for performing amyloid PET imaging in the management of AD, how to interpret the scans, and the limitations of the test. PET quantification methods used to assess Aβ plaque load in the brain will also be clarified.

## 2. The Role of PET Amyloid Imaging in the Management of AD

There are various radiotracers that have been experimented with to enable in vivo detection of amyloid depositions in the brain. Among the main indications for performing an amyloid scan is to confirm the diagnosis in indeterminate or atypical cases, to aid in early detection of AD among subjects with mild cognitive impairment (MCI) and to enable quantification of progressive changes in brain amyloid load in response to anti-amyloid therapy. Currently, there are several compounds detected to have a high affinity to bind with fibrillary Aβ plaques that have received FDA approval. There is also a newer benzofuran derivative for the imaging of Aβ in vivo, i.e., [18F]FPYBF-2 (5-(5-(2-(2-(2-18F-fluoroethoxy)ethoxy)ethoxy)benzofuran-2-yl)-N-methylpyridin-2-amine), which is chemically stable, does not photodegrade, and has been tested for feasibility in distinguishing AD patients [24].

### 2.1. [^11^C]Pittsburgh Compound-B

In January 2004, Klunk et al. published the first human amyloid PET study using ^11^C-labeled Pittsburgh Compound-B ([^11^C]PiB), also known as 2-(4-N-[^11^C]methylaminophenyl)-6-hydroxybenzothiazole. This compound, which is radiolabeled to ^11^C, has approximately 20-minutes half-life. Therefore, its use is restricted to centres that have an on-site cyclotron facility. Compared with healthy controls, AD patients typically showed marked retention of [^11^C]PiB in areas of cerebral cortex that have been identified to contain substantial amounts of deposition of fibrillar Aβ plaques in AD patients [25]. Interestingly, an inverse relationship was noted between [^11^C]PiB cortical retention with cerebral glucose metabolism determined with [^18^F]FDG, which was most prominent in the parietal lobes [25].

Using distribution volume ratio (DVR) method, Rowe et al. demonstrated that there was higher [^11^C]PiB binding in neocortical areas in AD and DLB patients when compared with healthy control subjects, and no cortical binding in FTD [26]. Although almost all AD patients have significantly positive [^11^C]PiB scans, especially in later stages of the disease, even healthy adults can have falsely positive scan results, e.g., up to 12% of adults in their 60s, 30% of adults in their 70s, and a minimum of 50% of adults in their 80s [26]. The annual incidence of healthy subjects to develop [^11^C]PiB positivity, as reported in a publication from the Mayo Clinic, evaluating elderly population, was estimated to be 13% per year [27].

### 2.2. [^18^F]Florbetapir Scan

[^18^F]Florbetapir ([^18^F]FBP), which is also known as ^18^F-AV-45 or 4-{(E)-2-[6-(2-{2-[2-(18F)Fluoroethoxy]ethoxy}ethoxy)-3-pyridinyl]vinyl}-N-methylaniline, is an ^18^F-labelled amyloid PET ligand that has high accuracy in detecting brain amyloid deposition [28]. It is taken up rapidly through the BBB, and rapidly washed out from grey matter tissue that does not contain amyloid. It has a high affinity to aggregated Aβ, good separation between the radiotracer amyloid retention and background signal, and a long, stable pseudo-equilibrium allowing flexibility in the timing of image acquisition [29]. [^18^F]FBP received the U.S. FDA approval in 2012, to be the initial nuclear medicine imaging radiotracer allowed to be utilized for subjects who are being investigated for AD or other causes of cognitive decline [30]. It has an advantage over [^11^C]PiB because it has a longer half-life, which is approximately 110 min that makes it possible to transport it from site of production to regional PET scanner facilities.

It also demonstrates excellent specificity to Aβ detection and has favorable pharmacokinetics, whereby it is quickly cleared from the blood circulation with approximately 10% of the radiotracer remaining at 20 min post injection. The radiotracer readily enters the brain and 20 min post injection, a clear separation can be seen between individuals with and without significant cerebral amyloid deposition (Figure 1).

[^18^F]FBP is maximally taken up in the brains of subjects with presumed presence of aggregated fibrillar Aβ after 30 min post injection. Furthermore, its biodistribution is stable for up to 60 min of being introduced intravenously. This allows a wide time window to facilitate the recommended imaging protocol of 10 min. Radiation dosimetry assessments in humans have indicated that the highest organ exposures to the introduced amyloid radioligand occur in the liver, gallbladder, urinary bladder, and the gut [28].

It is noteworthy that a significant amount of amyloid burden has been detected in [^18^F]FBP scans among even cognitively normal elderly patients who have uncontrolled hypertension and underlying risk factors such as the presence of one or more APOE ε4 alleles [31]. Nevertheless, [^18^F]FBP is widely used as a research biomarker in the Alzheimer’s Disease Neuroimaging Initiative (ADNI) project and in several phase III clinical trials of experimental AD drugs.

### 2.3. [^18^F ]Florbetaben

[^18^F]Florbetaben ([^18^F]FBB) is another radiotracer used in amyloid PET imaging. In a study carried out in 2012 by Villemagne et al. [^11^C]PiB and [^18^F]FBB images were co-registered so that the selection of regions of interest (ROIs) were similar in both scans [32]. This was followed by utilizing the cerebellar cortex as a region of reference to calculate the standard uptake value ratios (SUVr). This gave good concordance of elevated SUVr using [^18^F]FBB in subjects with AD compared to healthy controls [32].

### 2.4. [^18^F]Flutemetamol

[^18^F]Flutemetamol ([^18^F]FMT) is a PET radiotracer with an extended half-life, which enables it to be distributed in vivo for a longer duration compared to [^18^F]FBP. The data indicate that, as an amyloid imaging agent, the performance of [^18^F]FMT is similar to that of [^11^C]PiB in AD and AD-associated MCI [32]. Furthermore, a comparison made among amyloid radiotracers [^11^C]PiB, [^18^F]FBP, and [^18^F]FMT revealed good contrast between composite amyloid retention ratio that was normalized by whole cerebellum, along with consistent detection of Aβ positivity in the scans [33].

## 3. The Role of PET Using Tau Imaging Radioligands

PET imaging that detects tau proteins is able to differentiate AD from other types of NCDs, namely DLB. The PET ligand [^18^F]AV-1451 (formerly known as T807) binds to a broad spectrum of tau-positive inclusions, but not to tau-negative aggregates in autopsy samples from NCDs other than AD. In 2015, Rabinovici et al. reported preliminary in vivo results applying [^18^F]AV-1451 PET imaging to patients with NCDs other than AD, which were associated with positive or negative tau pathology with promising results [34]. Additionally, [^18^F]Florbenazine (^18^F-AV-133), which is a biomarker for vesicular monamine type 2 transporters (VMAT2), has been noted to be significantly decreased in the Parkinson’s disease/DLB group compared to AD patients, and can help to differentiate the two conditions in atypical cases. Neurocognitive impairment in DLB patients is significantly associated with VMAT2 density. Furthermore, significant differences have been detected in consecutive [^18^F]AV-133 and [18F]FBP scans that are consistent with expected NCD pathology, i.e., greater amyloid depositions in AD and DLB as compared with Parkinson’s disease and controls [35].

## 4. Enabling Early and Accurate Diagnosis of AD

In as much as AD progression is attributed to three cardinal neuropathological features, i.e., deposition of senile plaques in the extracellular matrix, neurofibrillary tangles (NFTs) accumulated in the intracellular compartment and synaptic degeneration, it is believed that it is actually the fibrillary Aβ particles that are crucial for the ultimate dysfunction of neurons and development of AD [36]. Although Aβ pathology is consistently detected in post mortem studies of AD patients, it has been reported to be present in 25–45% of normal elderly individuals [37]. This finding depends on the age of the cohort and the pathological criteria employed. It has been suggested that the presence of this pathology reflects the earliest stage in the development of AD [38]. Proving this hypothesis is a real challenge, caused by limitations inherent in post mortem analysis. Fortunately, recent developments of functional amyloid imaging that binds to fibrillar Aβ plaques, provides a unique opportunity to quantify this pathology in the ante mortem state.

[^18^F]FBP, [^18^F]FBB, and [^18^F]FMT all have good affinity to bind with Aβ plaques, and are potentially excellent non-invasive tools for in vivo imaging of NCDs. In familial hereditary AD, significant brain amyloid burden appears years before symptoms of NCD [39], plus accumulation of amyloid deposits may start earlier than in sporadic AD. Amyloid imaging, particularly using the cerebellar grey matter or cerebral grey matter, has the potential clinical utility for aiding in differential diagnosis in early-onset AD and to support the clinical diagnosis of subjects with probable AD with improved diagnostic accuracy [1,40,41,42,43,44] (Table 1).

Alzheimer’s Association and the Society of Nuclear Medicine and Molecular Imaging convened the Amyloid Imaging Taskforce (AIT) to gather empirical evidence that can help support the clinical utility of amyloid imaging. The AIT agreed upon a set of specific appropriate use criteria (AUC), ultimately empowering authorized medical practitioners to identify the types of patients and clinical circumstances in which amyloid PET could be used. It is recommended that patients who have persistent unexplained MCI, patients with atypical presentations of AD, and patients at an atypically early age who develop progressive NCD symptoms are more likely to benefit from this test [45]. As it is a very sub-specialized imaging tool, it is recommended that AD experts decide on the patients that would best benefit from an amyloid scan. The expert for this purpose should be self-identified as a physician trained and board-certified in neurology, psychiatry, or geriatric medicine who actively devotes a substantial proportion (≥25%) of patient contact time to the evaluation and care of adults with acquired mild cognitive impairment (MCI) or AD, as confirmed by peer recognition [45].

Although almost all AD patients have significantly positive amyloid scans, especially in the later stages of life, even cognitively normal elderly adults can have falsely positive scan results, which acts as a caveat for the interpretation of positive amyloid scans. Notably, the annual incidence of healthy subjects becoming amyloid scan positive, as reported in a recent study from the Mayo Clinic, evaluating elderly subjects, was estimated to be 13% per year [27].

## 5. Prognostication of Disease and Detecting Potential MCI Converters

In patients who are diagnosed with MCI, it may be beneficial to perform amyloid PET imaging to assess the plaque burden. This imaging has been known to detect high plaque burden, which can indicate a higher chance of the individual to convert to AD. Furthermore, many recent studies postulate that perhaps imaging is not a stand-alone test and the addition of CSF biomarkers, genetic information, and glucose metabolism to aid in the diagnosis may help improve the accuracy of interpretation of results [24,46,47,48,49] (Table 2). The role of amyloid PET imaging in MCI subjects has been extensively researched with variable results. Nevertheless, it is important to note that amyloid imaging is contraindicated in asymptomatic individuals because there are potentially more harmful than beneficial effects on the subject due to inaccurate assumptions made about the risks of developing AD and future outcomes on the basis of scan results per se.

## 6. Aiding in Treatment Planning and Monitoring

Amyloid PET neuroimaging can act as a surrogate biomarker for the assessment of neuronal function and integrity. It can be utilized to quantify the beta amyloid peptide plaque burden in the cortices of the brain [1]. The development of this radiotracer has enabled works related to the formulation of disease-modifying drugs that can dissolve the intracerebral amyloid plaques. Hence, this type of targeted neuroimaging can act as a quantitative biomarker for the assessment of treatment response.

## 7. Pharmacokinetics of Amyloid Tracers and Scanning Protocol

Amyloid radioligands demonstrate high affinity and specificity to Aβ and favorable pharmacokinetics. The tracer, particularly [^18^F]FBP (approximately 370 Mbq or 10 mci), is injected intravenously [1], followed by approximately 30–50 min uptake time, some radioligands such as [18F]FMT may require 90 min and others such as [18F]FBB may require up to 130 min to achieve equilibrium [40]. The image acquisition time is commonly 10 min. The compounds are rapidly cleared from circulation with only 10% remaining 20 min after injection. The substances readily enter the brain and 20 min post injection, a clear separation can be seen between individuals with and without amyloid (Figure 2, Figure 3 and Figure 4).

In brains presumed to have aggregated Aβ, maximum uptake of [^18^F]FBP occurs approximately 30 min after injection and remains essentially unchanged for the subsequent 60 min, thus providing a wide time window to facilitate the 10-minute imaging protocol. Whole-body radiation dosimetry studies in humans indicate the pathway for excretion of the radioligand, with the gallbladder, intestines, liver, and urinary bladder demonstrating the highest exposure [28].

## 8. In Vivo Assessment and Measurement of Cerebral Amyloid Burden

The clinical use of amyloid PET commonly relies on visual assessment and interpretation (VA) of scans, in which the signal intensity of “target-rich” brain regions such as the frontal cortex is contrasted to that of “target-poor” regions such as subcortical white matter (Figure 2). VA can be used for assessing the likelihood of significant fibrillar Aβ burden in the brain. A systematic qualitative VA of images is necessary, so that evidence of standardized interpretation protocols that lead to acceptable levels of inter-rater agreement can be achieved. The interpretation of amyloid scans needs to be conducted by an imaging expert, to determine the presence or absence of Aβ plaque deposition. The imaging expert is usually a nuclear medicine specialist or radiologist with specific training in the interpretation of amyloid PET [45]. The amyloid PET data must be technically adequate and must be acquired at a fully qualified and certified facility for it to be of diagnostic quality.

The criteria for interpreting the [^18^F]FBP scan as positive for AD by the VA method is identifying the presence of amyloid binding in the cortical regions relative to the cerebellum, which gives a poor grey-white matter contrast in the affected cerebral cortex. Among the brain areas affected by familial type of AD, which is related to the presence of amyloid precursor protein (APP) and Presenilin gene mutations, the striatal regions have been reported to have the highest amyloid depositions [50]. Sporadic type of AD that is related to APOE ε4 carrier status does not seem to have any regional predilection for amyloid burden; however, the amyloid deposition tends to have an earlier appearance in ApoE epsilon 4 carriers than non-carriers [50]. In a study performed by Newberg et al., [^18^F]FDG PET scans were interpreted as positive if they displayed the classic pattern of hypometabolism in the temporo-parietal regions [51]. Based on that criterion, scans were classified as either positive or negative for AD. In addition, relative scoring systems have been used to assess the degree of either FDG hypometabolism or increased amyloid binding characteristic of specified regions based on a priori knowledge. Cluster analysis by La Joie et al. revealed distinct subsets of regions [52]: in the hippocampus, atrophy exceeded hypometabolism, whereas Aβ load was minimal,in posterior association areas, Aβ deposition was predominant, together with high hypometabolism and lower but still significant atrophy, andin frontal regions, Aβ deposition was maximal, whereas detection of structural and metabolic alterations was low.

Atrophy and hypometabolism significantly correlated in the hippocampus and temporo-parietal cortex, whereas Aβ load was not significantly related to either atrophy or hypometabolism. Thus, they postulated that these findings probably reflect the differential involvement of region-specific pathological or protective mechanisms, such as the presence of neurofibrillary tangles (NFTs), disconnection, as well as compensation processes.

## 9. Quantitative Interpretation of Cortical Aβ

The quantitative assessment of amyloid plaques requires consistent efforts for standardization of protocols. The steps involved include proper subject selection and management, calibrated dosage of radiotracer administration, image analysis and quality control, selection of brain reference region, and optimization of various image processing and segmentation methods [50]. Technical factors can be attributed to lead to variability in measurements, and can affect interpretation of the scans, e.g., scanner parameters, injected dose, movement correction methods, and uptake time.

Among the regions of the brain that have the lowest fibrillary amyloid plaques are the cerebellar white matter and almost none in the cerebellar grey matter. Thus, these regions are preferentially used to calculate SUVr values. Particularly, the cerebellar cortex was selected as a reference standard for the very first clinical study using [^11^C]PiB PET due to its absence of Congo red and thioflavin-S stained plaques [51]. In addition, the clearance of amyloid radiotracers from cerebellar grey matter is considered more similar to its clearance from the cerebral grey matter target regions [52]. Similarly, Schmidt et al. described quantitative assessment of cortical amyloid signal (reflecting amyloid plaque deposition) relative to a reference region that is believed not to accumulate amyloid (normal reference area), producing the SUVr measurement [50]. Wong et al. employed parametric reference region method or a simplified SUVr calculated from 10 min of scanning, 50–60 min after [^18^F]FBP administration [29].

A step by step approach at assessment of amyloid scans is to scrutinize the CT scan images for abnormalities in the structure. Subsequently, ROIs will be selected to evaluate cortical regions associated with significant amyloid burden in MCI and AD patients. By using the Montreal Neurological Institute (MNI) atlas on SPM software the scans are normalized to the standard brain template and mean cortical and cerebellar values are calculated to give the SUVr data. Routinely, six (6) cortical grey matter regions are targeted for the ROI analysis, i.e., anterior and posterior cingulate, medial orbital frontal, temporal, and parietal lobes, and the precuneus [53].

There have been many preset threshold SUVr values that have been proposed to improve the sensitivity and specificity of amyloid PET scans (Table 1, column 8 and Table 2, column 9). Among the initially recommended preset cutoff points of SUVr ≥ 1.17 was indicated to reflect amyloid levels in the pathological range based on separate in vivo PET studies and autopsy reports from nineteen (19) end-of-life patients [54]. Hence, they were able to portray that [^18^F]FBP PET SUVr values can potentially be used to characterize fibrillary amyloid levels in clinical cases having probable AD, MCI, and among cognitively normal older adults, by using continuous and binary visual read measures of Aβ load. Frequently, concurrent MRI scans are used to normalize the PET data and improve on the anatomical localization of AD pathology.

Conversely, qualitative visual reads assessment using amyloid PET scans have been noted to have a sensitivity of approximately 84.6% (95% CI 0.55–0.98) and a specificity of approximately 38.1% (95% CI 0.18–0.62) for differentiating AD patients from healthy control subjects [55]. Nevertheless, improved scan parameters and the combination of other biomarkers have aided in improving the test accuracy (Table 1; Table 2). Although the quantitative assessment of the global cortex SUVr—despite being more time consuming—is advocated to be more accurate, with improved sensitivity of 92.3% and specificity of 90.5% using a threshold value of 1.122 (area under the curve, AUC 0.894) [55], much work is needed for the standardization of scanning and the reporting techniques, the protocol for global regional uptake assessment, and the selection of the region of reference.

In view of visual assessment and SUVr assessment having a discrepancy of approximately 10%, it is evident that we need to look elsewhere for further improvements in the diagnostic confidence of interpreting amyloid PET scans. Quantification is particularly useful in equivocal cases and for the purpose of providing numerical data that can reflect the change in amyloid load that is related to anti-amyloid therapy.

Finally, it is important to be aware that AD occurs as a spectrum of impairment in various levels of cognitive function, sometimes with additional confounding neuropsychiatric symptoms, which makes it difficult to have a standardized tool or threshold for confirming the diagnosis. As a matter of fact, it is well known that there are discordant areas of sites of atrophy noted on structural MRI, regional hypometabolism noted on ^18^F-FDG PET with areas of amyloid deposition detected by amyloid PET imaging. This phenomenon is likely due to the longitudinal evolution of hypometabolism, which occurs in the cerebral regions that are remotely located but connected by functionality with areas of increased amyloid burden [56]. Hence, it is evident that various imaging data, clinical, molecular, and genetic information are required to be evaluated in tandem as they provide complementary information for improved management of AD.

## 10. Limitations

The prevalence of amyloid PET positivity among healthy older adults is among the main limitations of this diagnostic imaging. It is speculated that age-specific positivity rates for amyloid PET are less than 5% in those 50–60 years old, 10–12% in those 60 to 70 years old, 25–30% in those 70–80 years old, and increases exponentially to more than 50% in persons aged 80–90 years [57].

Another major caveat is that a positive amyloid scan can also be seen not only in AD, but also in other medical conditions. For example, amyloid PET is frequently positive in DLB [58]. Nevertheless, [^18^F]FDG scans can play a role in identifying the ‘cingulate island sign’ (CIS), which is present in DLB and demonstrates increased glucose metabolism in the posterior cingulate cortex [59,60]. The CIS is highly specific for DLB (97–100%) but with lower sensitivity (62–86%) [60]. The sensitivity of detecting DLB by neuroimaging can be improved by presynaptic dopaminergic imaging using [^123^I]FP-CIT (DaTSCAN, GE Healthcare), which can give a pooled sensitivity of 86.5% (95% CI: 72–94.1%), but with relatively lower specificity of 93.6% (95% CI: 88.5–96.6%) [61]. Conversely, amyloid PET scans may, in certain instances, underestimate the brain amyloid plaque burden, especially in the setting of low cerebrospinal fluid (CSF) Aβ_42_ levels and mild NCD of the AD type [62]. In cases of FTD, which has a similar insidious onset as AD but with predominantly language and behavioral abnormalities, [^18^F]FDG PET may detect hypometabolism more frequent in the frontal, anterior cingulate, and anterior temporal regions, as opposed to the temporo-parietal and posterior cingulate regions in AD [63]. Although there can be some overlapping areas that demonstrate [^18^F]FDG, PET amyloid scans often can differentiate FTD from AD as these cases demonstrate significantly lower amyloid uptake in frontal, temporo-parietal, and occipital lobes as well as in the putamen [64].

## 11. Conclusion

The aim of performing imaging scans in AD is to non-invasively aid in the diagnosis and provide objective confirmatory evidence of the cause of the neurocognitive deficit. This also aids the management of cognitive impairment, which includes prophylactic planning to anticipate future requirements of the patient. Overall, at present, PET amyloid imaging may promise a beneficial role to diagnose AD in inconclusive cases; however, there is an inherent limitation namely in its cost effectiveness and practical concerns for its execution due to many variations in protocols and cutoff values for interpretation of results.

## 12. Key Points

Aβ deposition can be accurately detected by PET amyloid scans.AD subjects will usually have a positive amyloid scan.Caution needs to be exercised during interpretation and reporting of scan results, as positive amyloid scans can be seen in cognitively normal older adults, AD, and other subtypes of dementia such as DLB.The degree of amyloid deposition does not correlate with the severity of AD.Severity of amyloid deposition in young subjects (with increased genetic susceptibility) may be a prognostic factor for the development of early onset AD among them.Amyloid PET qualitative evaluation of cerebral amyloid presence can be made by binary visual assessment, with loss of grey white matter differentiation denoting a positive scan.Quantification of Aβ can be made using the whole cerebellum, cerebellar grey matter, and other regions of low to nil physiological amyloid deposition, as a reference point to calculate the standardised uptake value ratio (SUVr), using the assumption that these regions are usually spared in AD.

## 13. Recommendation

Future works should include recommendations for newer imaging radioligands, improved automated quantification of amyloid imaging, newer techniques for improved specificity in detecting subjects with Aβ deposition, and prognostication of at risk older adults for the possibility of developing Alzheimer’s disease.

## Figures and Tables

**Figure 1 diagnostics-09-00065-f001:**
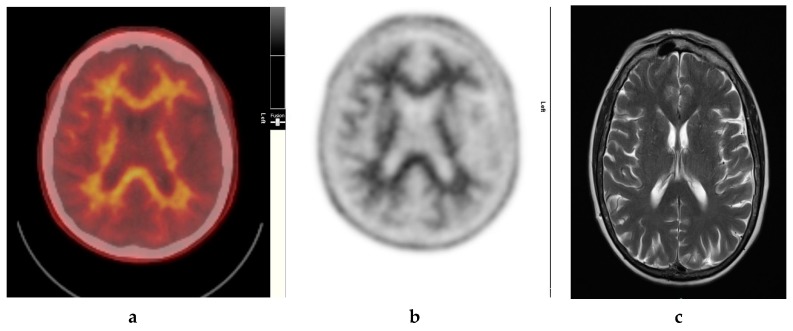
Normal study. Sixty-seven year old female patient with deteriorating cognition and multiple vascular risk factors for assessment for vascular dementia. (**a**) and (**b**) [18F]FBP shows good contrast between grey and white matter in all sections of the brain with no obvious evidence of beta amyloid plaque disease. This suggests that the diagnosis/development of Alzheimer’s disease is less likely. (**c**) MRI brain scan (multiplanar and multi sequence acquisitions) shows no significant T2 signal abnormality or restricted diffusion to suggest space occupying lesion, infarction, or ischemic change. (figures are courtesy RLBUHT Hospital database).

**Figure 2 diagnostics-09-00065-f002:**
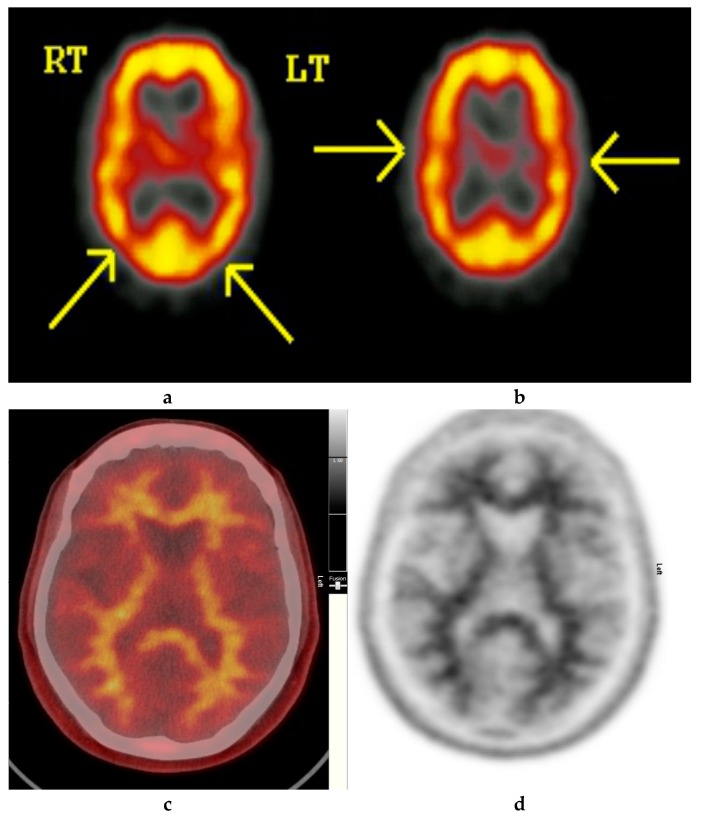
Normal study. Fifty-seven year old female patient memory problems/loss and confusion with a query of Alzheimer’s disease. (**a**) and (**b**) Technetium-99m HMPAO-SPECT brain Scan [^99m^Tc]HMPAO-SPECT brain scan. Brain scan showed mildly reduced perfusion to both temporal and both parietal lobes (yellow arrows) with relatively better perfusion anteriorly. Reduction of cerebral blood flow in these regions is commonly seen in patients with early Alzheimer’s dementia rather than vascular dementia. However, in view of the subtle nature of the appearances on the [^99m^Tc]HMPAO-SPECT brain scan, the [^18^F]FBP brain scan was performed to identify beta amyloid plaque disease and to give a higher confidence for AD diagnosis or suggest vascular aetiology. (**c**) Color coded [^18^F]FBP scan. (**d**) Grey scale [^18^F]FBP scan shows good contrast between grey and white matter in sections of the brain and no obvious evidence of beta amyloid plaque disease. This means that the likelihood of developing AD is low and as such the overall findings are more suggestive of vascular aetiology rather than early AD. (Figures are courtesy RLBUHT Hospital database).

**Figure 3 diagnostics-09-00065-f003:**
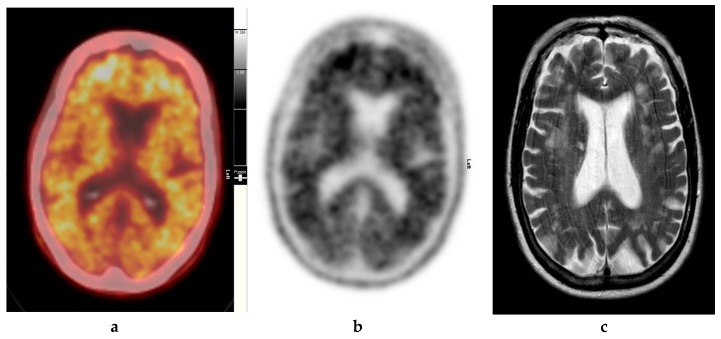
Abnormal study. Seventy-four year old male patient with a history of short term memory problems/gradual memory decline for some time and a query of dementing pathology. (**a**) and (**b**) [18F]FBP shows loss of contrast between grey and white matter in all sections of the brain. The scan is suggestive of beta-amyloid plaque deposition and in a patient with the above clinical presentation is suggestive of early AD. (**c**) MRI brain scan (multiplanar and multi sequence acquisitions) with some motion artefact shows generalised age appropriate cerebral atrophy, proportionate symmetrical temporal lobe atrophy, and corresponding dilatation of the cerebrospinal fluid (CSF) spaces. Moderate to marked periventricular T2 white matter hyperintensities likely to represent chronic small vessel ischemic changes. No diffusion restriction or space occupying lesions were identified. (Figures are courtesy RLBUHT Hospital database.).

**Figure 4 diagnostics-09-00065-f004:**
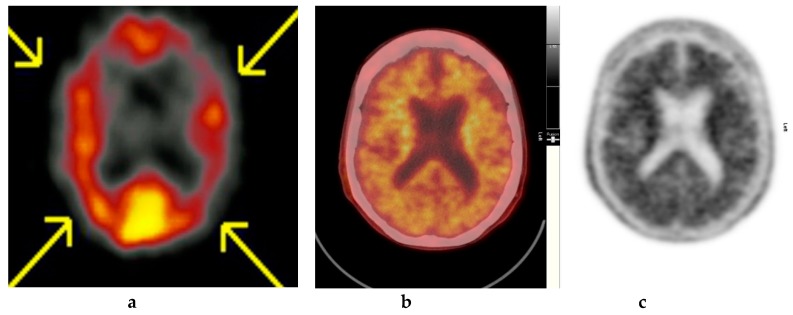
Abnormal study. Sixty-eight year old female patient with speech difficulties, deficits in memory, and visuospatial abilities. (**a**) [^99m^Tc]HMPAO-SPECT brain scan shows asymmetrical heterogenous reduced perfusion to both parietal and temporal lobes (worse on the left side) as with involvement of the frontal lobes identified by the yellow arrows. Normal areas of temporal perfusion were noted in between the abnormal parietal and frontal areas. In a patient with cognitive impairment, the findings indicate patterns of cerebral blood flow commonly seen in patients with mixed dementia, i.e., Alzheimer’s disease with vascular dementia. (**b**) Color coded [^18^F]FBP scan. (**c**) Grey scale [^18^F]FBP scans show loss of contrast between grey and white matter in all sections of the brain suggestive of beta-amyloid plaque deposition/disease. In conjunction with the [^99m^Tc]HMPAO-SPECT brain scan, the overall findings are again suggestive of vascular as well as beta amyloid plaque disease (early AD), making the diagnosis mixed type AD. (Figures are courtesy RLBUHT Hospital database).

**Table 1 diagnostics-09-00065-t001:** Clinical utility of amyloid positron emission tomography (PET) scans in diagnosing Alzheimer’s disease (AD).

Author (Year)	Amyloid Tracer	Subjects	Age	Dose of Tracer (MBq)	Uptake Time (min)	Clinical Ref.	Reference Standard	Findings
Chiang (2017) [40]	[^11^C]PiB	Cognitively healthy older adults	63 ± 5	555	60	DSM-5	MRS at hippocampus PiB: Cerebellar GM	Corrected for APOE ε3/ε4 positivity: ↓ glutathione levels associated with ↑ amyloid load at the hippocampus
Chiaravalloti (2018) [41]	[^18^F]FBB, [^18^F]FDG	AD: 38, Control FDG: 58	AD: 69 ± 8	[^18^F]FBB: 295–320, FDG: 185–210	[^18^F]FBB: 90	MMSE: 21.7 ± 5.9	ROI placed at selected cortical GM	FDG hypo-metabolism correlated with Amyloid positivity at temporal, parietal and limbic regions. Mean normalised SUVr: 1.28 +/− 0.1
Ciarmiello (2019) [42]	[^18^F]FBB	66 (MCI)	75.97 ± 6.59	306 ± 29	86 ± 8	MMSE 25.4 ± 3.07	cerebellar GM	SUVr 1.3 = positive scan, 54% of positive scans correlated with AD neuropathology among MCI
Miki (2017) [43]	[^18^F]FMT	Total: 70 AD: 25 MCI: 20 Controls: 25	75 ± 6	Single dose: 185 Cumulative dose: 240	90	NINCDS-ARDRA DSM-IV	Cerebellar GM	Visual reads: PPV: 88–92% NPV: 96–100%
Pothier (2019) [44]	[^18^F]FBP	Cognitively normal older adults: 65 (MCI: 31, Controls: 34)	76.11 years old ± 4.73	4 MBq/kg body weight	50	MMSE, CDR	cerebellar GM	visual reads: Aβ+ > 1.21, SUVr cutoffs: Aβ+ > 1.17, Aβ− < 1.10, equivocal amyloid load as in between range. no significant difference in cognitive decline in Aβ+ and Aβ− groups
Suppiah (2018) [1]	[^18^F]FBP	47 (Probable AD: 17, Possible AD: 30)	Probable AD: 63.5 ± 9.2, Possible AD: 62.7 ± 10.7	370	30	DSM-5, MMSE	visual reads of global cortical uptake compared with cerebellar GM	sensitivity: 62.5%, specificity: 77.4%, PPV: 58.8%, NPV:80.0%, severity of amyloid load was not correlated with diagnosis of probable AD

[^18^F]FBB: Florbetaben, APOE: apolipoprotein genotype, AD: Alzheimer’s disease, MCI: mild cognitive impairment (particularly amnestic type), CDR: Clinical Dementia Rating scale, Aβ+: amyloid beta positive PET scans, Aβ−: amyloid beta negative PET scans, NINCDS-ADRDA: National Institute of Neurological and Communicative Disorders and Stroke and the Alzheimer ’s disease and Related Disorders Association, WM: white matter, GM: grey matter, Aβ42/40: CSF amyloid beta 42/40 ratio, ROI: region of interest, SUVr: standardized uptake value ratio, ref.: reference, DSM: Diagnostic and Statistical Manual, PPV: positive predictive value, NPV: negative predictive value, MRS: MRI spectroscopy.

**Table 2 diagnostics-09-00065-t002:** Amyloid PET findings correlated with other biomarkers of AD and mild cognitive impairment (MCI).

Author (Year)	Amyloid Tracer	Other Biomarkers	Subjects	Age (Years)	Dose of Tracer (MBq)	Uptake Time (min)	Clinical Reference	Reference Standard	Findings
Alongi (2019) [46]	[^18^F]FBB	CSF amyloid levels	44 (neuro-cognitive deficit)	AD: 72.3, Controls: 68	269 ± 10%	90	ENS-EFNS criteria, MMSE	cerebellar WM	SUVr highest at precuneus, Amyloid PET: sens: 90.9% spec: 78.9%, CSF amyloid < 750 pg/mL: sens: 81.5% spec: 75.9%
Bouter (2019) [47]	[^18^F]FBB	CSF amyloid levels	33 (neuro-cognitive deficit)	68.4 ± 10.3	300	90	MMSE 25.2 ± 3.0	global cortex	↓Aβ42/40 & SUVr correlated with MMSE, mean SUVr: APOE ε4 carrier: 1.489, non-carriers: 1.313
Frings (2018) [48]	[^11^C]PiB	[^18^F]FDG	39 (MCI)	Converter: 69.8 ± 7.1 Non converter: 70.0 ± 6.4	[^11^C]PiB: 393 ± 56 [^18^F] FDG: 224 ± 36	[^18^F]FDG: 50–70	NINCDS-ADRDA	cerebellar GM	[^11^C]PiB PET predicted conversion from MCI to AD, HR for positive [^11^C]PiB scan: 10.2 (95% CI 1.3–78.1)
Higashi (2018) [24]	[^18^F]FPYBF-2	[^11^C]PiB	Controls: 61, Cases with suspected AD: 55 (AD: 27, MCI: 16, CN: 3, other NCDs: 9)	Controls: 53.7 ± 13.1, Cases: 74.4 ± 9.4	200 ± 22	50–70	DSM-IV and DSM-5 NINCDS-ADRDA	cerebellar GM	good correlation of PiB with ^18^F-FPYBF-2, Mean Cortical Index: early AD: 1.288 ± 0.134 moderate AD: 1.342 ± 0.191, PiB SUVr: 1.435 ± 0.474
Kim (2018) [49]	[^18^F]FBB, [^18^F]FMT	APOE	523 (MCI) Aβ+: 238 Aβ−: 285	Validation Set Aβ+: 71.4 ± 7.2 Aβ-: 69.7 ± 8.2	[^18^F]FBB: 311.5 [^18^F]FMT: 197.7	90	DSM-IV and DSM-5, Seoul Neuro-psycho-logical Screening Battery	Visuals reads based on uptake at selected ROI	Positivity for APOE ε4 (OR 4.14) among MCI is associated with PET Aβ+.

[^18^F]FBB: Florbetaben, [^18^F]FMT: [^18^F]-Flutemetamol, APOE: apolipoprotein genotype, AD: Alzheimer’s disease, MCI: mild cognitive impairment (particularly amnestic type), CN: cognitively normal by neurophychiatric tests, Aβ+: amyloid beta positive PET scans, Aβ−: amyloid beta negative PET scans, ENS-EFNS: European Federation of the Neurological Societies dementia Guidelines, NINCDS-ADRDA: National Institute of Neurological and Communicative Disorders and Stroke and the Alzheimer ’s disease and Related Disorders Association, WM: white matter, GM: grey matter, Aβ42/40: CSF amyloid beta 42/40 ratio, ROI: region of interest, SUVr: standardized uptake value ratio, HR: hazard ratio, [^18^F]FDG: ^18^F Fluorodeoxyglucose, DSM: Diagnostic and Statistical Manual, [^18^F]FPYBF-2: 5-(5-(2-(2-(2-^18^F-fluoroethoxy)ethoxy)ethoxy)benzofuran-2-yl)-N-methylpyridin-2-amine.

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
