# Peer review of "The Who, When, Why, and How of PET Amyloid Imaging in Management of Alzheimer’s Disease—Review of Literature and Interesting Images"

_diagnostics, 2019, doi:10.3390/diagnostics9020065_

Round 1

Reviewer 1 Report

Major comments:

The author should unify format of the mass number both in text and in table, such as “18F-FDG,” or “11C-PiB,” not “F18-FDG” nor “C11-PiB.” The mass number such as “18” and “11” should be a superscript.

Similarly, the terms "PETCT" and "PET / CT" are used in combination. In this review, I think that the term "PET" is simply good.

In the “Introduction and role of 18F-FDG PET/CT,” the authors describes the differentiation of AD from other diseases. However, the differential diagnosis using 18F-FDG PET was not described. AD can be denied in negative amyloid images. Amyloid images can not distinguish between AD and DLB. The amyloid images may be positive in normal subjects or in FTLD. Authors should describe the features of FDG PET in each NCD, such as cingulate island sign in DLB.

The authors described “Previous research has focused on 18F-FDG PET/CT studies as it is a widely available radiotracer for various imaging indications and has well established cutoff points for standardized uptake values (SUVs) utilized for disease process quantification.” While 18F-FDG PET for oncology frequently uses SUV, FDG PET use cerebral metabolism ratio of glucose (CMRgluc) instead of SUV in the brain. Alternatively, a relative image with a cerebellum or the like as a reference area is used.

Minor comments:

Page 1, line 37;  Introduction and role of F18-FDG PETCT
–>”Introduction and role of
18F-FDG PET/CT,”

Page 3, line 111;  subtypes of neurocognitive disorders (NCDs)
 –> subtypes of NCDs

Page 13, Figure 4(a); The SPECT image is collapsed vertically.

Author Response

Comments to both reviewers and editor's own comments attached for your information

Reviewer 2 Report

Diagnostics_522613_peer_review

Subapriya Suppiah, Mellanie-Anne Didier and Sobhan Vinjamuri

The Who, When, Why and How of PET/CT Amyloid Imaging in Management of Alzheimer’s Disease—Review of Literature and Interesting Images

The authors aimed “to expound on the various arrays of indications for performing amyloid PET/CT imaging, and to interpret the scans and discuss the limitations of the method. PET/CT quantification methods used to assess Aβ plaque load in the brain was also clarified.”

It is obvious that the authors have wide experience in clinical PET imaging especially of neurocognitive disorders.

However I have several suggestions that should be taken into account before this manuscript can be published.

My comments:

1.     The nomenclature of presented radiopharmaceuticals should follow the guidelines presented in:

“Consensus nomenclature rules for radiopharmaceutical chemistry — Setting the record straight, Heinz H.Coenen, Antony D.Gee, MichaelAdam, GunnarAntoni, Cathy S.Cutler, YasuhisaFujibayashi, Jae MinJeong, Robert H.Mach, Thomas L.Mindt, Victor W.Pike, Albert D.Windhorst, published in Nucl Med Biol, 2017, https://doi.org/10.1016/j.nucmedbio.2017.09.004”. It is strongly recommend that all manuscripts meet these guidelines upon submission.

These recommendations have also been taken in use e.g. in all EJNMMI journals.

This means that the tracer names should be written: [18F]FDG, [18F]Flobetapir, [18F]Florbetaben, [18F]Flutemetamol, [11C]PIB, [99m]Tc. I.e. the used radionuclide in square brackets and 18, 11, 99m etc. as superscripts. But without square brackets if e.g. 18F-labelled.

Please correct ALL tracer names. Also those used in tables and figure legends.

2.     [18F]Flutemetamol is misspelled in Tables 1 and 2. and females and males should be written with small initials in Figure legends.

3.     Please correct the misspelled name of [18F]Florbetapir in the Figure 1 legend.

4.      [18F]FPYBF-2 is mentioned in the Table 2 but not elsewhere. Please either remove or add to the text.

5.     The abstract is not in line with the contests of the manuscript. I.e. “Additionally, some insights on the development of newer radiotracers and recommendations for future developments in this field are also provided”. This part should be presented more clearly because it is difficult to find among the text. Please correct the abstract so that it matches the manuscript text itself.

6.     Keywords: Please check if the keywords are the best ones.

7.     The full chemical name has been added to the [18F]Florbetapir but not to the other amyloid tracers. Please add the chemical names also to [18F]FDG, [18F]Florbetaben, [18F]Flutemetamol, and [11C]PIB. Please use the IUPAC-name if possible e.g. 2-Deoxy-2-[18F]fluoroglucose.

8.     The shorter name of FBB has been used for [18F]Florbetaben The correct acronym is [18F]FBB. If the authors decide to use the acronym it should be used throughout the manuscript after it has been mentioned. The same concerns the acronym [18F]AV-45 for [18F]Florbetapir. Not to use them both.

9.     Figures 1, 2, 3 and 4 are missing the references. Are the figures from some previously published articles or from some researchers? The references should be mentioned.

10.  The chapter 2 describes “The role of PET/CT Amyloid imaging in the management of AD”. Its sub-chapter 2.5. “PET/CT using tau imaging radioligands” does not fit under the same name. If the authors want to say something about tau-PET-imaging it should be under a separate chapter and they should explain why it is important to discuss tau-PET imaging under this title.

11.  Limitations: Some of the limitation have been discussed under the tracer itself. It would be better if all limitations are described under the same subheading. Here it could be nice to expand this chapter e.g. describing with a few words how frontotemporal dementia (FTD) cannot be distinguished from Alzheimer´s disease using amyloid imaging but using [18F]FDG-PET it is possible. Lewy body disease (DLB) shows also high uptake with in amyloid PET but can be distinguished from AD using dopamine transporter imaging either with SPECT or PET.

12.  References: ALL references should be written as stated in the Instructions for Authors:

Author1, A.B.; Author2, C.D. Title of the article. Abbreviated Journal Name. Year, Volume, page range. Abbreviated Journal name can be found from PubMed.

E.g. Ref. 1 should be: Suppiah, S.; Ching, S.M.; Nordin, A.J.; Vinjamuri, S. The role of PET/CT amyloid Imaging compared with Tc99m-HMPAO SPECT imaging for diagnosing Alzheimer's disease. Med J Malaysia. 2018, 73(3), 141-146.

and

Ref. 2: Dhikav, V.; Sethi, M.; Anand, K.S. Medial temporal lobe atrophy in Alzheimer's disease/mild cognitive impairment with depression. Br J Radiol. 2014, 87(1042), 20140150.

Please correct ALL references and check also that all titles are written exactly like in the original paper.

Author Response

(The authors gave the same response as above.)
